

# Reflective properties of white and snow-covered sea ice

Aleksey Malinka[1], Eleonora Zege[1], Georg Heygster[2], Larysa Istomina[2]

[1]Institute of Physics, National Academy of Sciences of Belarus, 220072, pr. Nezavisimosti 68, Minsk, Belarus
[2]Institute of Environmental Physics, University of Bremen, O. Hahn Allee 1, D-28359 Bremen, Germany

5    *Correspondence to*: Aleksey Malinka (mal@light.basnet.by)

**Abstract.** White ice (ice with a highly scattering granular layer on top of its surface) and snow-covered ice occupy a large part of the sea ice area in the Arctic, the former in summer, the latter in the cold period. The inherent optical properties (IOPs) and the reflectance of these types of ice are considered from the point of view of the light scattering and radiative transfer theories. The IOPs – the extinction and absorption coefficients and the scattering phase function – are derived in the assumption that both the snow cover and the scattering layer of white ice are random mixtures of air and ice with the characteristic grain size significantly larger than the wavelength of incident light. Simple analytical formulas are put forward to calculate the bidirectional reflectance factor (BRF), albedo at direct incidence (the directional–hemispherical reflectance), and albedo at diffuse incidence (the bihemispherical reflectance). The developed optical model is verified with the data of the *in situ* measurements made during the RV *Polarstern* expedition ARK-XXVII/3 in 2012.

## 1 Introduction

Ice cover is the main factor governing the radiative budget in the Arctic (Curry et al., 1995; Eicken et al., 2004; Køltzow, 2007; Pirazzini, 2008; Shindell and Faluvegi, 2009; Serreze et al., 2011). Monitoring its state, including remote sensing, is of great importance, especially in times of the strong environmental changes we see nowadays (Serreze et al., 2000; Dethloff et al., 2006; Perovich et al., 2008; Pistone et al., 2014). The changes in the Arctic sea ice are particularly noticeable in summertime, when solar light controls the processes of sea ice transformation. Also it is the season when satellite optical sensors are able to deliver information about polar regions. Our goal here is to develop a physical model of the reflective properties of summer ice, which is essential for the development of satellite remote sensing methods, as well as for the correct interpretation of the results of field measurements for understanding the physics of ice.

There is a great variety of the sea ice types (Bogorodskii, 1970; Byshuev et al., 1974; Untersteiner, 1990; Perovich et al., 2009; Nicolaus et al., 2010; Sea Ice Nomenclature, 2014). 'White ice' is not a strict term from the nomenclature; rather it puts together the various types of ice with high albedo and got its name due to its white appearance, produced by a highly scattering top layer (Grenfell and Maykut, 1977; Perovich, 1996). This layer is formed after melt water has drained off from the surface elevations into the depressions. It consists of ice grains with the order of millimeters in size and, thus, can be described within the same approach as snow, but with larger grains (see Fig. 1). This layer provides stable high reflectance and, in fact, determines the reflective properties of white ice (Grenfell and Maykut, 1977; Perovich, 1996; Perovich et al.,

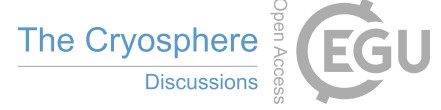



2002; Herzfeld et al., 2006). White ice and snow are the brightest surfaces in the Arctic, which occupy a large part of the sea ice area, the former in the period of melting, the latter in the cold period (Grenfell and Maykut, 1977; Perovich et al., 2002). That is why it is so important to characterize accurately their reflective properties, especially with regard to monitoring the ice field albedo and melting processes from optical satellite observations (Herzfeld et al., 2006; Tschudi et al., 2008; Rösel et

al., 2012; Warren, 2013; Zege et al., 2015).

## 2 Inherent optical properties

An optical model of any medium is specified on the base of some microphysical model. The common way to model the reflection of white ice and snow is to interpret them as consisting of independent spherical scatterers, the Mie solution being often used to calculate the IOPs (Bohren and Barkstrom, 1974; Choudhury and Chang, 1979; Wiscombe and Warren, 1980;

Grenfell et al., 1994; Light et al., 1998; Grenfell and Warren, 1999; Hamre et al., 2004). However, a set of spherical scatterers, even polydispersed, has a number of typical scattering features, such as a minimum at the scattering angle of about $100^0$, a rainbow, or a glory. Neither white ice nor snow demonstrate these scattering features, so the Henyey-Greenstein scattering function is often used instead of the Mie one in the radiative transfer simulations (Light et al., 1998; Grenfell and Warren, 1999; Aoki et al., 2000; Hamre et al., 2004). At the same time, both optical and microphysical

measurements show that the ice grains in snow and in the granular layer of white ice are really far from spheres and, in fact, irregularly shaped (Massom et al., 2001; Kokhanovsky et al., 2005; Matzl et al., 2006; Kerbrat et al., 2008; Domine et al., 2008; Picard et al., 2009). Kokhanovsky and Zege (2004) developed the model of light scattering in snow that uses just the fact that the grain size is much larger than the wavelength and that the imaginary part of the refractive index is small. This model matches the case of white ice, because the substance is the same (ice) and the typical grain size is even larger.

Although the model was successfully used in many applications, e.g., in the retrieval of the snow grain size from satellite observations (Zege et al., 2011; Wiebe et al., 2013), there is still some dissatisfaction caused by the phenomenological nature of the model. The more consistent way is to consider snow (and hence white ice) as a porous material and to apply to its description the model of a random mixture based on the stereological approach (Malinka, 2014). This approach uses the concept of the chord length distribution rather than the characteristics of a separate particle and only requires that the mixture

is stochastic. The mean chord length equals the mean photon path length inside one of the components of the mixture and plays the role of the effective size of a grain or a gap. In addition to the requirements of stochasticity, the model of a random mixture (Malinka, 2014) uses the laws of geometrical optics. It is appropriate to recall that geometrical optics is applicable if the characteristic obstacle size is much larger than the wavelength, the real part of the refractive index differs significantly from unity, and its imaginary part is small. A typical grain of fresh snow is about 100 μm (and grows when getting older)

and, as it was mentioned above, a granule in white ice is of the order of millimeters, which is evidently much larger than the wavelength of visible and IR light. The spectral behavior of the complex refractive index of ice measured by Warren and Brandt (2008) is shown in Fig. 2.





As it is seen from Fig. 2, the refractive index of ice meets the above mentioned requirements in the wavelength range of $0.3 - 2$ μm.

## 2.1 Microphysical characteristics

Most microphysical characteristics of the random mixture can be expressed in terms of the mean chords $a$ and $h$ that are the mean photon path length in ice and air gaps, respectively (Kendall and Moran, 1963; Pielou, 1964; Switzer, 1965; Gille, 2000; Gille et al., 2002). Thus, the volume fraction of ice $\beta$, the porosity $\phi$, the bulk density $\rho$, and the ice density $\rho_{ice}$ are related by:

$$\beta = 1 - \phi = \frac{\rho}{\rho_{ice}} = \frac{a}{a+h}. \tag{1}$$

The autocorrelation length $l_c$ is:

$$l_c = \frac{ah}{a+h}. \tag{2}$$

The specific area $s$ of the ice-air interface per unit volume of the mixture is:

$$s = \frac{4\phi(1-\phi)}{l_c} = \frac{4}{a+h}. \tag{3}$$

The specific surface area (SSA) per sample mass equals:

$$SSA = \frac{s}{\rho} = \frac{4}{(a+h)\rho_{ice}\beta} = \frac{4}{\rho_{ice}a}. \tag{4}$$

Relationship (4) is valid not only for the model of a porous material, but also for an ensemble of randomly oriented convex particles, which is a limit case of a random mixture. In that case the mean chord $a$ is inversely related to the commonly used surface-area-to-volume ratio $\langle S \rangle / \langle V \rangle$:

$$a = \frac{4\langle V \rangle}{\langle S \rangle} = \frac{\langle V \rangle}{\langle S_\perp \rangle}, \tag{5}$$

where $S_\perp$ is the particle projection area, and symbol $\langle \ \rangle$ denotes ensemble averaging.

Equation (5) indicates that the mean chord in the ensemble of particles coincides, to an accuracy of a multiplier, with the standard definitions of the effective size of irregular grains in snow (Kokhanovsky and Zege, 2004; Domine et al., 2008; Zege et al. 2008; Picard et al., 2009) and the effective radius of polydispersed spherical droplets in aerosols and clouds (Dobbins and Jizmagian, 1966; Naumenko, 1971; Hansen and Travis, 1974). However, Eq. (4) is more general, as the model of the stochastic mixture does not include the requirement of grain convexity and can be applied to a dense-packed medium. On the other hand, the coincidence of the mean chord with the standard definition of the effective particle size opens the physical meaning of the latter: it equals the mean photon path length inside a particle.





## 2.2 Light scattering characteristics

The optical properties of the random mixture are determined by the spectral behavior of the complex refractive index $m+i\kappa$ and the mean chords $a$ and $h$. The main light scattering characteristics used in the radiative transfer theory are the extinction coefficient $\varepsilon$, the photon survival probability (single scattering albedo) $\omega_0$, and the scattering phase function $p(\theta)$.

For the random mixture, to which the laws of geometrical optics are applicable, these values are equal (Malinka, 2014) to:

$$\varepsilon = \frac{1}{a+h} ,\tag{6}$$

$$\omega_0 = 1 - \frac{xT_{diff}}{x+T_{diff}} ,\tag{7}$$

with

$$x = \alpha n^2 a ,\tag{8}$$

where $\alpha$ is the absorption coefficient of ice:

$$\alpha = \frac{4\pi}{\lambda}\kappa ,\tag{9}$$

and $T_{diff}$ is the Fresnel transmittance of diffuse light through the air-ice boundary:

$$T_{diff} = \frac{2(5n^6+8n^5+6n^4-5n^3-n-1)}{3(n^3+n^2+n+1)(n^4-1)} + \frac{n^2(n^2-1)^2}{(n^2+1)^3}\ln\frac{n+1}{n-1} - \frac{8n^4(n^4+1)}{(n^4-1)^2(n^2+1)}\ln n .\tag{10}$$

In the spectral range 0.3-1.1 µm (the range we will consider hereinafter) the real part $n$ of the refractive index of ice changes in the range 1.300-1.334; the value $1-T_{diff}$ changes in the interval $6.11\times10^{-2} \div 6.95\times10^{-2}$.

The expression for the phase function is a little bit more complex. It is presented as a series in the Legendre polynomials $P_l(x)$:

$$p(\theta) = \frac{1}{\omega_0}\left[R_{out}(\theta_i) + \frac{1}{n^2}\sum_{l=0}^{\infty}(2l+1)\frac{t_l^2}{1+\alpha a - r_l^{in}}P_l(\cos\theta)\right].\tag{11}$$

Here, $R_{out}(\theta_i)$ is the Fresnel reflectance of the air-ice interface for incident angle $\theta_i$; $t_l$ and $r_l^{in}$ are the coefficients of expansion of the functions $F_t(\theta)$ and $F_{in}(\theta)$ in the Legendre polynomials:

$$F_t(\theta) = \sum_{l=0}^{\infty}\frac{2l+1}{4\pi}t_l P_l(\cos\theta), \qquad F_{in}(\theta) = \sum_{l=0}^{\infty}\frac{2l+1}{4\pi}r_l^{in}P_l(\cos\theta) ,\tag{12}$$

where the functions $F_t(\theta)$ and $F_{in}(\theta)$ describe the angular distribution of light, transmitted and internally reflected by the air-ice interface, respectively:





$$F_t(\theta) = \begin{cases} T_{out}(\theta_i)\dfrac{n^2(n\cos\theta-1)(n-\cos\theta)}{\pi(n^2-2n\cos\theta+1)^2}, & \cos\theta>1/n \\ 0, & \cos\theta<1/n \end{cases}$$

$$F_{in}(\theta) = \frac{R_{in}(\theta_{in})}{4\pi},$$

(13)

where $T_{out}(\theta_i)$ and $R_{in}(\theta_{in})$ are the Fresnel transmittance at incidence angle $\theta_i$ and the internal reflectance at incidence angle $\theta_{in}$. Angles $\theta$, $\theta_i$, and $\theta_{in}$ are related as:

$$\theta = \theta_i - \arcsin\frac{\sin\theta_i}{n}, \qquad \theta_{in} = \frac{\pi-\theta}{2}.$$

(14)

From Eqs. (11)-(14) it is possible to derive the analytical expression for the average cosine $g$ of the phase function:

$$g = \langle\cos\theta\rangle = \frac{1}{2}\int_0^\pi p(\theta)\cos\theta\sin\theta\,d\theta$$

$$= \frac{1}{\omega_0}\left(r_1 + \frac{n^2 t_1^2}{T_{diff}(1-n^2)-r_1+n^4(1+\alpha a)}\right),$$

(15)

where

$$r_1 = \frac{n(3n^{11}+3n^{10}+25n^9+25n^8+22n^7-282n^6+138n^5+186n^4+151n^3-89n^2+13n-3)}{24(n+1)(n^4-1)(n^2+1)^2}$$
$$+\frac{8n^4(n^6-3n^4+n^2-1)}{(n^4-1)^2(n^2+1)^2}\ln n - \frac{(n^8+12n^6+54n^4-4n^2+1)(n^2-1)^2}{16(n^2+1)^4}\ln\frac{n+1}{n-1},$$

$$t_1 = \frac{3n^8+3n^7-17n^6+55n^5-39n^4-7n^3-27n^2-11n-8}{24(n+1)(n^4-1)n}$$
$$-\frac{(n^2-1)^4}{16(n^2+1)^2 n}\ln\frac{n+1}{n-1}+\frac{4n^5}{(n^4-1)^2}\ln n.$$

(16)

The example of the spectral dependence of the photon survival probability $\omega_0$ and the average cosine $g$ for the 'ice-air' random mixture with $a=2\,mm$ is shown in Fig. 3. It is seen that the medium is practically non-absorbing in the visible and near IR range (the photon arrival probability is greater than 0.9 in the interval 0.3-1.1 µm), which justifies the name of 'white ice'. The average cosine $g$ takes the values from 0.63 at 0.3 µm to 0.69 at 1.1 µm for $a=2\,mm$, with the mean value of about 0.67.

The phase functions of the same mixture at different wavelengths are shown in Fig. 4. The wavelengths are chosen to be at the edges of the visible range (380 and 700 nm) and in SWIR (2 µm). The phase functions are similar in the forward scattering region for all the wavelengths and are practically independent of wavelength in the visible range.





## 2.3 Absorption

The spectral absorption of the random mixture is mainly determined by the imaginary part of the refractive index (see Eqs. (7)-(9)). In the case of ice it has a pronounced minimum at about $0.3 - 0.4$ µm (see Fig. 2). However, the spectral behavior of white ice reflectance measured *in situ* shows in many cases the decrease in the blue range (see Figures below). This effect can be explained by the measurement geometry, when light from the sky and direct sunlight are added in different proportions at different wavelengths (see Sect. 4.2). However, the careful analysis of the albedo field measurements, as well as the satellite data, shows that this effect is not sufficient to explain this spectral behavior. This means the presence of some absorbing contaminant.

There can be different contaminants in ice: the organic pigments from sea water, the particles of sediments from the atmosphere, which could be both long-distance transferred (as the Sahara dust) or local (pollution from industrial centers). E.g., the clay, slit, and sand particles are found in the ice situated far from a coastline in the Beaufort Sea (Reimnitz et al., 1993) and in the Central Artic (Nürnberg et al., 1994). However, at our sight, ideally suited to the role of ice contaminant absorbing in the blue region is the yellow substance: dissolved organic matter (DOM) from the sea water. The spectrum of the yellow substance (Bricaud et al., 1981) with the corrections done by Kopelevich et al. (1989) is:

$$\alpha_y(\lambda) = \begin{cases} \alpha_y(390)\exp\left[-0.015(\lambda - 390)\right], & \lambda \leq 500, \\ \alpha_y(390)\exp\left[-0.015(500 - 390) - 0.011(\lambda - 500)\right], & \lambda > 500, \end{cases} \tag{17}$$

where $\lambda$ is in nanometers and $\alpha_y(390)$ is the only free parameter that determines the spectrum.

Here the concentration of DOM is represented implicitly through its absorption coefficient $\alpha_y(\lambda)$ at $\lambda = 390nm$. This is a conventional way to describe the DOM absorption used in the ocean optics (see, e.g., the fundamental monograph of Mobley, 1994).

To take into account the absorption by the yellow substance in white ice we rewrite Eq. (9) as:

$$\alpha = \frac{4\pi}{\lambda}\kappa + \alpha_y . \tag{18}$$

The presence of the yellow substance in both Arctic and Antarctic ice was reported by many authors (Thomas et al., 1995; Thomas et al., 2001; Bhatia et al., 2010; Norman et al., 2011; Beine et al., 2012; Grannas et al., 2014; Hansell and Carlson, 2014;) and was sufficient to represent the shape of most of the reflection spectra we have analyzed. Once the more detailed data on the ice contaminants are available, their absorption spectra can be easily incorporated into the model through Eq. (18).

## 3 Reflectance of white ice

The IOPs of the upper layer of white ice are defined in Sect. 2. The question that remains open is the stratification of IOPs in the lower layers. The IOPs of ice change drastically with depth; however the upper scattering layer is the main factor that





determines the white ice reflection. The lower layers may only slightly affect the reflectance spectrum. Moreover, the upper part of an ice sheet, just below the scattering layer, contains a lot of air inclusions (about 4-5%) (Gavrilo and Gaitskhoki, 1970; Mobley et al., 1998), thus its spectral behavior should be similar to that of the upper layer, which is a mixture of ice and air. Therefore, we pretend to describe the reflective properties of the whole ice sheet, considering just the scattering

layer, but with effective parameters that can slightly differ from the real ones to take into account the effect of the lower layers.

### 3.1 Asymptotic formulas

Given the inherent optical properties of a layer, the apparent optical properties (AOPs) can be calculated with any appropriate radiative transfer code. However, keeping in mind further application of the developed model to an inverse

problem, one would prefer to speed up the process of solution of the direct problem. Analytical formulas are most suitable for this purpose. As white ice and snow are bright surfaces with high albedo, the asymptotic formulas of the radiative transfer theory that describe the behavior of AOPs of an optically thick layer with weak absorption (Rozenberg, 1963; Germogenova, 1963; Hulst, 1968; Sobolev, 1975; Zege et al., 1991) can be applied to this case.

Here we consider the following spectral AOPs: the BRF (bidirectional reflectance factor) $R$, the albedo at direct incidence

$r(\theta_0)$, a.k.a. the directional–hemispherical reflectance, and the albedo at diffuse incidence $r_d$, a.k.a. the bihemispherical reflectance. The BRF depends on the polar angles of incidence and observation ($\theta_0$ and $\theta$, respectively) and the azimuth $\varphi$. Also all the AOPs are spectrally dependent. We omit the variables $\theta$, $\theta_0$, $\varphi$, and $\lambda$ for brevity, where it does not cause misunderstanding.

The following approximate solution $R_\infty$ for the BRF of a semi-infinite layer is given by Zege et al. (1991):

$$R_\infty = R_\infty^0 e^{-Y}, \qquad (19)$$

where $R_\infty^0$ is the BRF of the semi-infinite layer with the same scattering phase function, but with no absorption (i.e., for $\omega_0 = 1$), and the exponential factor describes the effect of absorption:

$$Y = y \frac{G(\theta)G(\theta_0)}{R_\infty^0},$$

$$G(\theta) = \frac{3}{7}(1+2\cos\theta), \qquad (20)$$

$$y = 4\sqrt{\frac{(1-\omega_0)}{3(1-\omega_0 g)}},$$

where $\omega_0$ and $g$ are, as usual, the single scattering albedo and the mean cosine of the scattering phase function.

For the albedos at direct and diffuse incidence Zege et al. (1991) suggest the appropriate formulas:





$$r(\theta_0) = \exp\left(-y\, G(\theta_0)\right),$$
$$r_d = e^{-y}.$$
(21)

The traditional asymptotic formulas of the radiative transfer theory for weak absorption are restricted by the first term of the expansion in the small parameter $y$ (Sobolev, 1975):

$$R_\infty = R_\infty^0 - y\, G(\theta) G(\theta_0),$$
$$r(\theta_0) = 1 - y\, G(\theta_0),$$
$$r_d = 1 - y.$$
(22)

However, the wider range of the absorption values is required for spectroscopy of scattering media for many problems as in remote sensing of ice and snow. Equations (19)-(21) are converted to the strict asymptotic ones (22) at small absorption ( $y \ll 1$ ), but they have a wider range of applicability, because they regard the terms up to the 3$^{rd}$ order of $y$ (Rozenberg, 1963).

Equations (19)-(21) were successfully used in remote sensing of snow (Zege et al., 2011; Wiebe et al., 2013). Their

efficiency is caused by the fact that the albedo of snow cover is very high (up to ~1 in the blue-green range), so that a layer of a dozen centimeters thick can be considered as optically infinite. Unlike a snow cover, white ice has the albedo of about 0.7-0.8 (or even less) in the blue-green region, what means that its optical thickness $\tau$ is finite. As a consequence, the optical thickness $\tau$ is the main parameter that determines its reflection and transmission.

The transition from the infinite to the large but finite optical depth $\tau$ can be made in the case of weak absorption by the

asymptotic formula (Germogenova, 1963; Hulst, 1968):

$$R = R_\infty - 2\frac{y\, G(\theta) G(\theta_0)}{e^{2Z} - 1},$$
(23)

where

$$Z = \gamma\tau + y,$$
(24)

$\gamma$ is the asymptotic attenuation coefficient:

$$\gamma = \sqrt{3(1-\omega_0)(1-\omega_0 g)}.$$
(25)

Using Eqs. (19) and (20), we get for Eq. (23):

$$R = R_\infty^0 e^{-Y} - 2\frac{y\, G(\theta) G(\theta_0)}{e^{2Z} - 1} = R_\infty^0\left(e^{-Y} - 2\frac{Y}{e^{2Z} - 1}\right).$$
(26)

In order to extend the scope of applicability of Eq. (26), we proceed in the spirit of Rozenberg (1963) and replace the linear term $Y$ by the hyperbolic term $\sinh Y$ :




$$R = R_\infty^0 \left( e^{-Y} - 2\frac{\sinh Y}{e^{2Z}-1} \right) = R_\infty^0 \left( e^{-Y} - \frac{e^Y - e^{-Y}}{e^{2Z}-1} \right)$$

$$= R_\infty^0 \frac{e^{-Y+2Z} - e^{-Y} - e^Y + e^{-Y}}{e^{2Z}-1} = R_\infty^0 \frac{e^{-Y+2Z} - e^Y}{e^{2Z}-1} = R_\infty^0 \frac{\sinh(Z-Y)}{\sinh Z}. \tag{27}$$

Finally,

$$R = R_\infty^0 \frac{\sinh\left(\gamma\tau + y\left[1 - G(\theta)G(\theta_0)/R_\infty^0\right]\right)}{\sinh\left(\gamma\tau + y\right)}. \tag{28}$$

For the albedos we have analogously:

$$r(\theta_0) = \frac{\sinh\left(\gamma\tau + y\left[1 - G(\theta_0)\right]\right)}{\sinh\left(\gamma\tau + y\right)},$$

$$r_d = \frac{\sinh\gamma\tau}{\sinh\left(\gamma\tau + y\right)}. \tag{29}$$

Formulas (29) coincide with those given in (Zege et al., 1991). Formulas (28)-(29) turn to formulas (19) and (21) for the semi-infinite layer at $\tau \to \infty$.

### 3.2 Numerical verification

Formula (28) was verified numerically with the radiative transfer code RAY (Tynes et al., 2001). Figures 5-6 present the BRF of white ice with $\tau = 8.5$ and $a = 3.333\,mm$ (these parameters correspond to the spectrum of the 'standard' white ice from (Perovich et al., 2002) at the wavelength of 490 nm (a non-absorbing layer) and 885 nm (a layer with significant absorption).

For the case of significant absorption the error of formula (28) is not greater than 5% if either $\theta$ or $\theta_0$ is less than $45^0$, and is not greater than 10% if both $\theta$ and $\theta_0$ are greater than $45^0$. For the non-absorbing layer the maximum error of formula (28) is 4% for both $\theta$ and $\theta_0$ equal to $0^0$.

Figure 7 presents the spectral albedo for the same layer. The error of the asymptotic formulas (29) is not greater than 2.5% for direct incidence and less than 1% for diffuse one.

Let us underline that formulas (28)-(29) are approximate. Their accuracy decreases with increasing absorption and with decreasing optical thickness. All-in-all: the brighter the layer the higher the accuracy.

Whereas the use of analytical formulas speeds up calculations extremely, as compared to numerical methods, these formulas will be useful in problems where the calculation time is crucial, i.e., in iterative loops.





### 3.3 Model outline

Let us summarize the above results. In the presented model the BRF of the semi-infinite non-absorbing layer $R_\infty^0$ and the mean cosine of the scattering phase function $g$ are determined by the phase function only and do practically not depend on wavelength in the visible range (in fact, in the range 0.3-1.1 µm, where the photon survival probability $\omega_0$ is close to 1).

Function $R_\infty^0$ is calculated only once with a radiative transfer code for different values of $(\theta, \theta_0, \varphi)$ and then used as a look-up-table; the value of $g \approx 0.67$ (see Fig. 3). The spectral dependence of reflectance is determined by the single scattering albedo $\omega_0$, which depends on the complex refractive index of ice $m$ and the effective grain size $a$. Absorption in the blue range can be affected by the adsorbed yellow substance (other possible pollutants are not taken into account, but can be easily included into the model).

Finally, the white ice reflective properties in the spectral range of 0.3-1.1 µm are determined by only three independent parameters given in Table 1.

These three values (together with the refractive index) completely determine the entire spectrum of all the reflective characteristics of white ice.

In the green range of the spectrum (~550 nm) both ice and yellow substance absorption is negligible. In the non-absorbing

case ($\omega_0 = 1$, $y = 0$, $\gamma = 0$), with the value of $g \simeq 0.67$, Eqs. (28)-(29) turn to the limiting form:

$$R = R_\infty^0 - \frac{4G(\theta)G(\theta_0)}{\tau + 4},$$

$$r(\theta_0) = 1 - \frac{4G(\theta_0)}{\tau + 4}, \qquad (30)$$

$$r_d = \frac{\tau}{\tau + 4}.$$

This form (30) can be useful for the estimation of the optical thickness of white ice by the reflection at its spectral maximum at about 550 nm.

### 4 Verification with the field data

Now there are several questions to be answered, concerning the developed model.

1. How reliable is the model? I.e., how close are the theoretical spectra determined with only few parameters, shown in Table 1, to the measured spectra of white ice at different situations?

2. What is the scope of applicability of the model? Can it be applied to the situations where the surface differs from the standard white ice?

3. What is the statistics of the ice parameters? What is the range of their changes in reality?



The last question is very important for the regularization of the inverse problem (retrieving the ice properties from optical data). When a problem is mathematically incorrect and needs regularization, as in this case, it is very important to have some *a priori* information, e.g., the range of variation of the sought-for parameters, to exclude false solutions.

### 4.1 Albedo measurements

The reflectance spectra of white ice were systematically measured for various melting ice situations during the RV *Polarstern* cruise ARK-XXVII/3, August 2 – October 8, 2012 (Istomina et al., 2013). The cruise track mainly followed the ice edge and only in the second half of the cruise the vessel entered thick multi-year ice. The thickness of the first-year ice at the edge varied from 0.5 m to 2-3 m (single floes and ridges), with the average of around 1-1.5 m (the data stem from visual estimation of overturning crashed ice floes done from the bridge of the vessel). During the cruise a variety of field conditions could be observed, including clear and cloudy skies of various cloud coverage, melting, freezing, and snowfall events. The sea ice surface featured a variety of crystal sizes, from very fine fresh fallen snow to ice granules of about 2-3mm. The measurements were done during the ice stations, when the vessel was parked at the ice floe for several days. Altogether there were nine ice stations, where the spectral albedo was measured (at any sky conditions). The measurements were always done around the solar noon at the given longitude. The measurement run took 2-3 hours, during which the solar angle, cloud coverage and surface conditions could change.

The portable spectroradiometer used for these measurements was the ASD FieldSpec Pro III, which uses three different sensors to obtain a spectrum from 350 nm to 2500 nm with the spectral resolution of 1.0 nm. The optical fiber cable used as a sensor was aimed at a $10 \times 10 cm^2$ Spectralon white plate, which was directed towards the measured surface and then towards the sky. The ratio of these two measurements of the upwelling and downwelling radiances gives the albedo of the surface. The sensor was held on a 1 meter long arm horizontally and facing perpendicularly to the surface (a bull's eye spirit level was used). The sensor was held above the surface at approximately 1 meter height. The measurements were taken each 10 meters along 200 m transect lines. A photograph of each measurement site was taken and a short description of the surface was documented.

Low sun elevation is responsible for the increase of noise in the measured spectra, especially in the middle-IR part. Therefore only measurements from the first six ice stations in the visible and near-IR range (0.35-1.35 μm) are used in this study. More details on the cruise can be found in (Boetius et al., 2013).

### 4.2 Measurements geometry and sky conditions

The spectral albedo measurements at natural illumination can be affected by different factors, such as imperfect sky conditions, e.g., overcast sky, scattered cumulus clouds, and thin cirrus clouds in otherwise clear skies. In the case of scattered clouds, especially when a cumulus cloud could obscure the solar disc during the measurement process, the change of sky conditions would inevitably distort the measured quantity and lead to unreasonable values of albedo. Such spectra were filtered out manually.



The measurements were carried out from the end of August to the beginning of September. For the days of measurements at the six stations the maximum solar elevation angles (at noon) were $21^0$, $20^0$, $19^0$, $17^0$, $15^0$, and $10^0$; i.e., the measurements were carried out at very low sun (zenith angle equals $90^0$ minus elevation angle). When the sun is low the direct solar flux is comparable to the diffuse flux from the sky, making the measured albedo value be a mixture of those at direct and diffuse

incidences. As the white ice albedo at direct incidence increases when the sun is approaching the horizon (see Eq. (30)), for the low sun the albedo at direct incidence is greater than that at diffuse incidence (the zenith angle, at which they are equal, is approximately equal to $\arccos(2/3) \simeq 48^0$, see Eq. (30)). It means that the light scattering by the sky makes the measured albedo lower than that at direct incidence. The greater the flux from the sky, the more essential this underestimation. In the clear sky dominating is the Rayleigh scattering, which has the strong spectral dependence, being significant in the blue and

negligible in the red and IR range. This leads to the situation when the measured albedo values are closer to that at diffuse incidence in the blue range and to that at direct incidence in the red, producing the reflectance decrease in the blue, which can be mistaken for the presence of a contaminant. The situation becomes even more complicated when dealing with such a bright surface as white ice, because of the essential multiple reflections between the surface and the atmosphere. To interpret the measurements data correctly we should examine this effect with more care.

Accurate documentation of atmospheric conditions in the field, including the atmosphere optical thickness, is needed in order to correct the field data, but is seldom available. In the *Polarstern* cruise during the measurements the following sky conditions were reported: the complete overcast, the overcast when the solar disk is visible, and the clear sky with scattered thin cirrus clouds.

In complete overcast conditions, when the solar disk is completely hidden, the contribution of the direct solar light is equal to

zero and the incident illumination is completely due to the diffuse sky light, so the measured albedo is the bihemispherical reflectance $r_d$.

Let us estimate the contribution of the direct light when the solar disk is visually observed through the clouds. Assuming approximately that the direct light brightness $B_0$ is constant across the solar disk, we can write in the case of overcast sky:

$$B_0 = \frac{E_0 e^{-\tau_{cl}/\cos\theta_0}}{\pi\gamma^2}, \tag{31}$$

where $E_0$ is the extraterrestrial solar flux, $\tau_{cl}$ is the optical thickness of the cloudy atmosphere ($e^{-\tau_{cl}/\cos\theta_0}$ is the attenuation factor), $\pi\gamma^2$ is the solid angle of the solar disk ($2\gamma \simeq 32'$).

When the solar disk is visible, the sky brightness $B_s$ is comparable in magnitude to the sun brightness:

$$B_s \sim B_0. \tag{32}$$

At the same time their fluxes are of different orders: the flux of the sky light is $\pi B_s$, while the solar flux is $E_0 \cos\theta_0 e^{-\tau_{cl}/\cos\theta_0}$.

Taking into account Eq. (32), we can say that the solar flux at the surface is less than the sky light flux by a factor of



$\gamma^{-2}\sec\theta_0$, i.e., by more than $4.6\times10^4$ times. This means that in this case the measured albedo also corresponds to that at diffuse incidence $r_d$.

Finally, the third case is the clear sky. Thin cirrus clouds may present but the sky is visually blue. The measured albedo corresponds to the value:

$$A=\frac{F_\uparrow}{F_\downarrow}, \tag{33}$$

where $F_\uparrow$ and $F_\downarrow$ are up- and downwelling fluxes, respectively, which in this case can be calculated approximately by the following formulas (Malinka et al., submitted):

$$F_\uparrow=\frac{RT}{1-r^a r_d},$$
$$F_\downarrow=T+\frac{r^a}{1-r^a r_d}RT, \tag{34}$$

where

$$RT=t_0 r(\theta_0)+t_d\frac{1+6r_d}{7}. \tag{35}$$

Here, as before, $r(\theta_0)$ and $r_d$ are the surface albedos at direct and diffuse incidence, respectively, $T=t_0+t_d$ is the atmospheric transmittance that is a sum of the direct solar light transmittance $t_0$ (attenuation factor) and the diffuse transmittance $t_d$ that describes the scattering in the atmosphere and, thus, determines the sky brightness; $r^a$ is the atmospheric albedo at diffuse incidence from below (Eqs. (34)-(35) account for the multiple reflections between the surface and the atmosphere).

So, while processing the field data (see the next Section), we interpret the measured value as the albedo at diffuse incidence in the cases of overcast and as given by Eqs. (33)-(35) for the clear sky. In the latter case the atmosphere model includes the molecular atmosphere (the Rayleigh scattering) with the Arctic Background aerosol (Tomasi et al., 2007) and a thin cirrus cloud layer with the optical depth of 0.1.

**4.3 Measured spectra retrieval**

Every spectrum consists of 1000 points of the measured values from 0.35 μm to 1.35 μm. These points were approximated with a curve, calculated by Eq. (29) (or Eq. (33) in appropriate cases) by fitting the three parameters from Table 1 (the best fit is understood in the least squares sense). Figures 8-12 present the comparison of the measured and retrieved albedo spectra. The respective retrieved ice parameters are presented in Table 2. A photo of the object of measurement is given for every plot. The measurement date and event number are shown (see Boetius et al., 2012 for details), as well as the sky conditions.





Figure 8 represents the bright surfaces. The typical grain size in these cases varies from 400 μm to 800 μm. This is an intermediate case between the typical snow and the typical white ice. The surface could be treated as aged snow when snow grains become larger. These cases clearly demonstrate that there is no strict separation between snow and white ice. Note that the first two spectra are affected by the yellow substance (decrease in the blue range), while the third one represents the

pure ice surface.

Figure 9 shows the typical white ice with the grain size of 1-4 mm, pure in the first two cases and with a little effect of the yellow substance in the third one. The discrepancies between measured and retrieved spectra at $\lambda > 1.2\,\mu m$ hereinafter are explained by the error of approximate formulas (29) for very low reflection.

Figure 10 presents the case of melt ice. The surface is dark and water saturated. The layer is characterized by low values of

the optical thickness (5 and lower) and high values of the effectives grain size (from 4 mm to 1 cm). This case is important, because it does not belong to the initial scope of the developed model, which is not presumed to describe wet snow or ice. Nevertheless, the spectra are retrieved satisfactory even when the spectral albedo maximum value is less than 0.4.

Snow-covered surfaces are presented in Fig. 11. The optical thickness is high (30 and higher), the grain size is about 100-300 μm, which is typical for fresh snow. The first spectrum is affected by a substantial amount of the yellow substance.

Overall, Figures 8 to 11 demonstrate that snow and white ice can be described in the framework of a uniform approach.

In Fig. 12 some outstanding cases are presented. The first one is crusted snow, which has a similar spectrum as typical white ice and, as a consequence, is described by similar parameters. The second one is a snow covered frozen pond. This case demonstrates that even a thin layer of snow has the optical thickness as an ordinary layer of white ice (~9). The large value of the effective grain size (~10 mm) does not match real snow grains, but rather it is responsible for high absorption in the IR

by the homogeneous ice under the snow layer. Apparently the same situation takes place in the third case, a frozen ice crack. Bubbles in the upper ice layer produce the situation, similar to the ice-air random mixture in the white ice scattering layer, but with a very large effective grain (~12 mm). Visually the crack looks dark and its total optical thickness is low (~2).

In general, the analysis of the experimental data shows that the developed model describes excellently the reflective properties of white ice and snow, at least as regards the spectral albedo, and quite satisfactory the cases that stand out of the

initial frames of the model such as wet ice/snow, crusted snow, frozen cracks, and snow covered ponds.

**5 Conclusion**

This work presents the optical model of white ice, i.e., any kind of ice covered with the scattering layer that consists of ice grains mixed with air. The main characteristics that determine its optical, particularly reflective, properties are the optical thickness and the effective grain size (the mean chord of the ice component in the random ice-air mixture). The model

considers only one pollutant, namely, the yellow substance that is responsible for absorption in the blue range. However, scattering and absorption by any sediment can be easily incorporated into the described model.



In addition, simple approximate formulas are put forward to calculate the BRF and albedo of a scattering layer of large but finite optical depth and low absorption, which is a case in point when dealing with white ice.

The verification with field data has shown that the model is sufficiently reliable: most of the measured spectra are retrieved with a high degree of accuracy by fitting only few parameters. The analysis has also shown that the model works quite satisfactory in the cases that stand out of the initial frames of the model such as wet ice/snow, crusted snow, frozen cracks, and snow covered ponds.

The statistical analysis of the measured data shows that the ordinary bare white ice has the optical thickness from 7 to 15 and the effective grain of 1-4 mm. However, the surface of white ice can be brighter, with unlimited optical thickness and smaller grains (from 400 μm to 1 mm). This surface is apparently a transition stage from fresh snow to an aged granular layer. Melting, water saturated ice forms a dark layer with the optical thickness of less than 5 and the effective grain size of 4 mm and larger (up to 10 mm). Fresh snow has the optical thickness usually greater than 30 and the effective grain size less than 300 μm. All of these surfaces can contain some amount of the yellow substance (DOM from the sea water); however, in the case of fresh snow the possible pollutant can likely be from another source, such as an atmospheric aerosol, and the experimental evidence of the yellow substance presence in white ice may be a matter for further investigations.

The presented model has been successfully used in the retrieval of the sea ice albedo and melt pond fraction from satellite optical data (Zege et al., 2015; Istomina et al., 2015a; Istomina et al., 2015b). As a whole, the presented approach is going to be useful for developing various retrieval techniques of satellite remote sensing, for studying underwater light fields, and for the problems of physics of sea ice and marine biology of the Arctic Ocean.

**Acknowledgements**

The work was supported by the 7[th] Framework Programme for Research and Development (project SIDARUS), financed by the European Union, and by the Junior Research Group for Remote Sensing of Sea Ice as a part of the Institutional Strategy of the University of Bremen, funded by the German Excellence Initiative.

The authors are grateful to the scientific party of the ARK XVII/3 cruise for making the spectral albedo measurements possible. Special thanks to D. Perovich for providing the ASD FieldSpec Pro III, M. Nicolaus for organizing the logistics, and the Sea Ice Physics group onboard for assisting with the measurements.

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



**Table 1. Characteristics of white ice that determine its reflective properties.**

| Symbol | Characteristics | Comments |
|---|---|---|
| $\tau$ | Effective optical thickness of the scattering layer | Main value that determines reflection in total and the only value that determines reflection in the range 500-550 nm, where absorption is absent |
| $a$ | Effective grain size (the mean chord) of the scattering layer (in μm) | Together with the complex refractive index determines the spectral behaviour of the reflectance |
| $\alpha_y(390)$ | Absorption coefficient of the yellow substance @ 390 nm (in m$^{-1}$) | Responsible for absorption in the blue range |



**Table 2. Retrieved parameters of white ice. See Table 1 for explanation of parameters.**

| Figure | Type | Plot | $\tau$ | $a$ (μm) | $\alpha_y$ (m$^{-1}$) |
|---|---|---|---|---|---|
| 8 | Bright white ice | 1 | $5.0\times10^2$ | $4.9\times10^2$ | $7.1\times10^{-1}$ |
| | | 2 | $3.2\times10^1$ | $4.5\times10^2$ | $2.0\times10^0$ |
| | | 3 | $1.4\times10^1$ | $7.2\times10^2$ | $2.9\times10^{-2}$ |
| 9 | Typical white ice | 1 | $9.3\times10^0$ | $2.8\times10^3$ | $6.7\times10^{-4}$ |
| | | 2 | $2.0\times10^1$ | $2.3\times10^3$ | $2.2\times10^{-4}$ |
| | | 3 | $1.9\times10^1$ | $2.2\times10^3$ | $1.9\times10^{-1}$ |
| 10 | Melting ice | 1 | $5.4\times10^0$ | $4.7\times10^3$ | $2.5\times10^{-4}$ |
| | | 2 | $3.4\times10^0$ | $6.7\times10^3$ | $6.0\times10^{-3}$ |
| | | 3 | $2.2\times10^0$ | $1.0\times10^4$ | $6.9\times10^{-3}$ |
| 11 | Snow-covered ice | 1 | $7.3\times10^1$ | $1.7\times10^2$ | $7.4\times10^0$ |
| | | 2 | $3.2\times10^1$ | $2.1\times10^2$ | $7.6\times10^{-3}$ |
| | | 3 | $2.7\times10^1$ | $2.7\times10^2$ | $1.0\times10^{-3}$ |
| 12 | Snow crust | 1 | $2.8\times10^1$ | $1.2\times10^3$ | $1.8\times10^{-1}$ |
| | Snow-covered pond | 2 | $8.7\times10^0$ | $9.4\times10^3$ | $6.6\times10^{-7}$ |
| | Crack | 3 | $2.3\times10^0$ | $1.2\times10^4$ | $4.4\times10^{-2}$ |



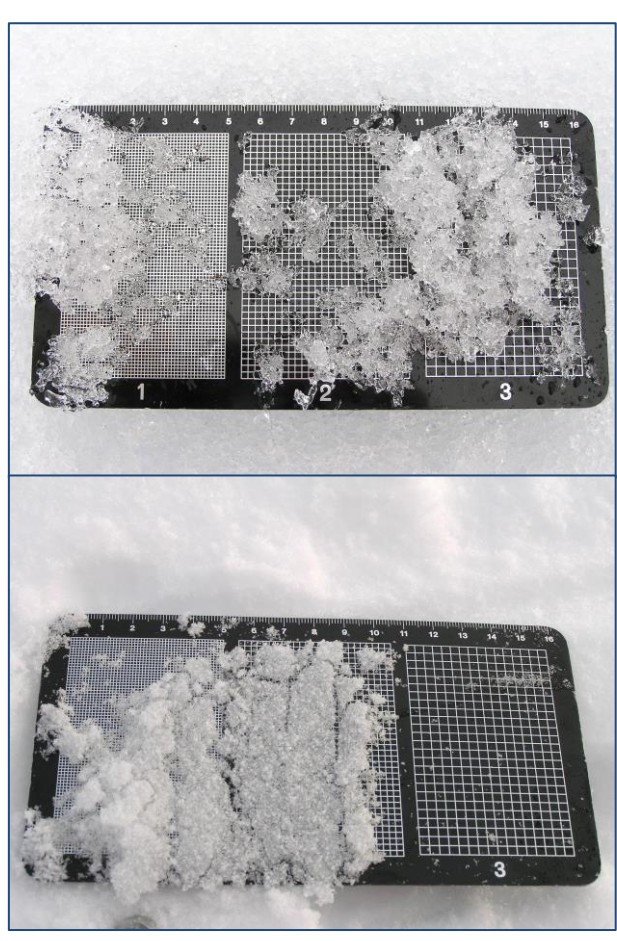

**Figure 1. Typical grains of white ice (top) and fresh snow (bottom) observed in the Central Arctic during ARK XXVII/3 in 2012. The size of the underlying grid is 1, 2 and 3 mm, respectively.**



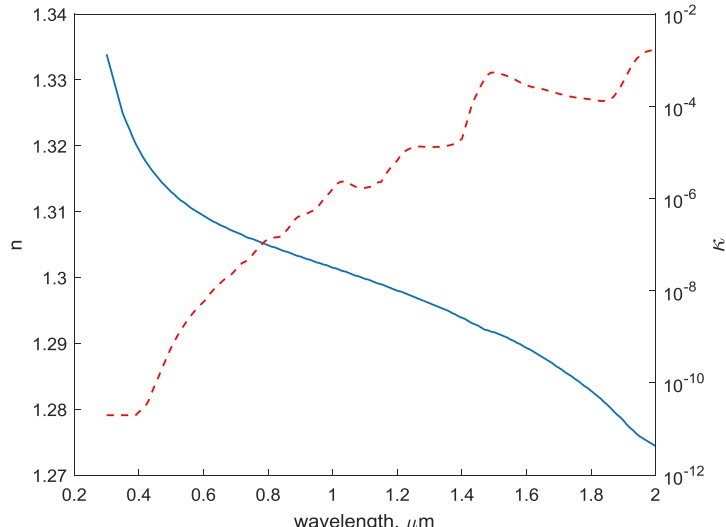

**Figure 2. The refractive index of ice after Warren and Brand (2008): real part (left axis, solid curve) and imaginary part (right axis, dashed curve).**



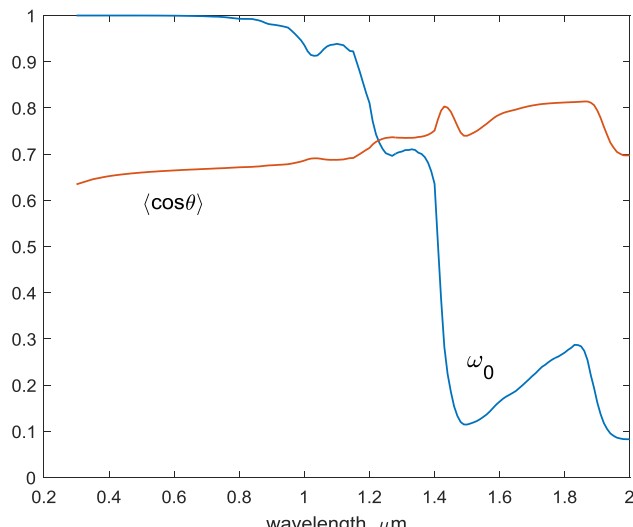

**Figure 3. Spectral dependence of the photon survival probability and the average cosine for the random mixture 'ice-air' with the mean chord of 2 mm.**





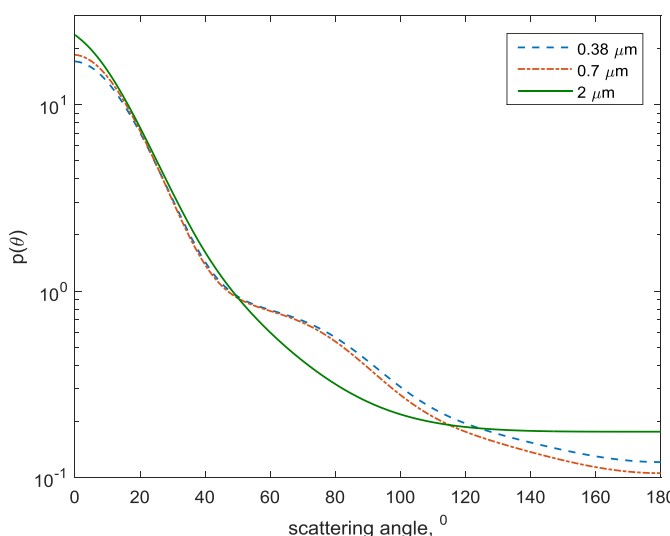

**Figure 4.** Simulated phase functions of the random mixture 'ice-air' with the mean chord of 2 mm for different wavelengths.





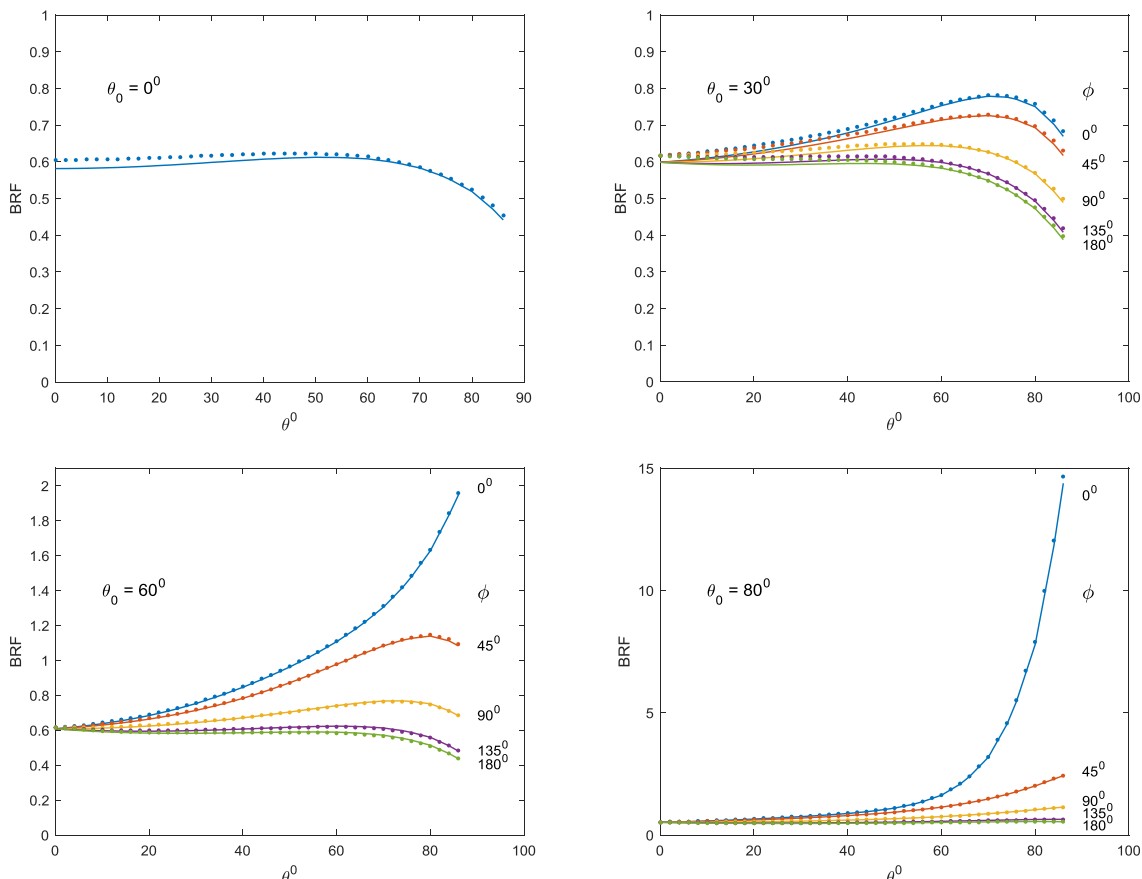

**Figure 5. BRF of white ice at 490 nm, calculated with the RAY code (dots) and the asymptotic formula (curves).**





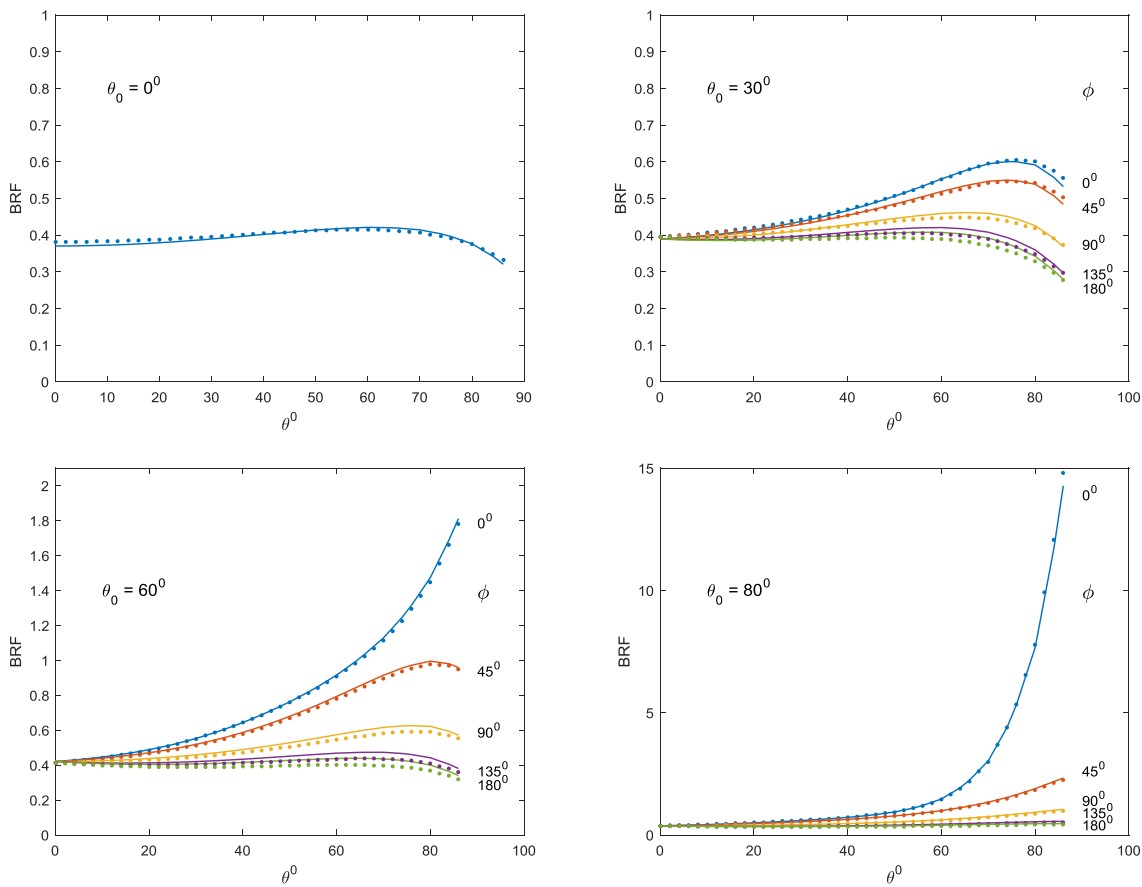

**Figure 6.** BRF of white ice at 885 nm, calculated with the RAY code (dots) and the asymptotic formula (curves).





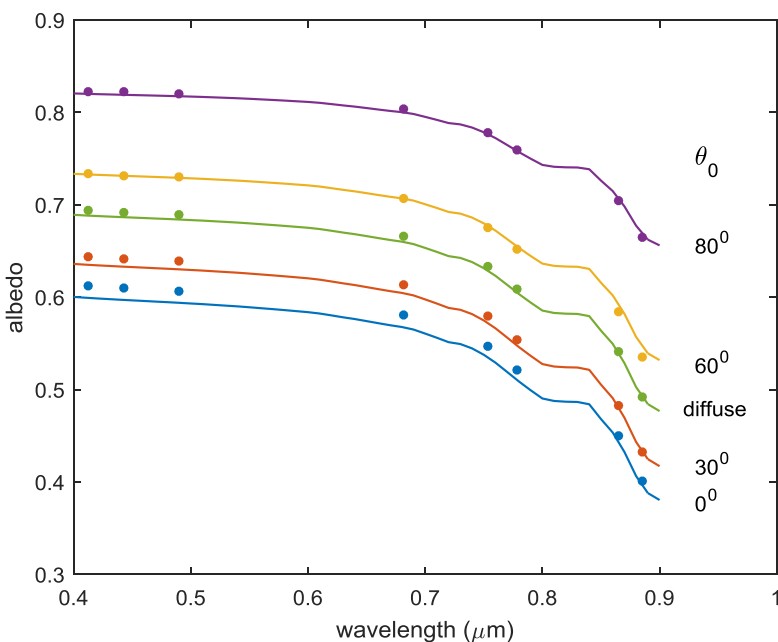

**Figure 7. Spectral albedo of the same layer at different incidence (direct and diffuse), calculated with the RAY code (dots) and the asymptotic formulas (curves).**



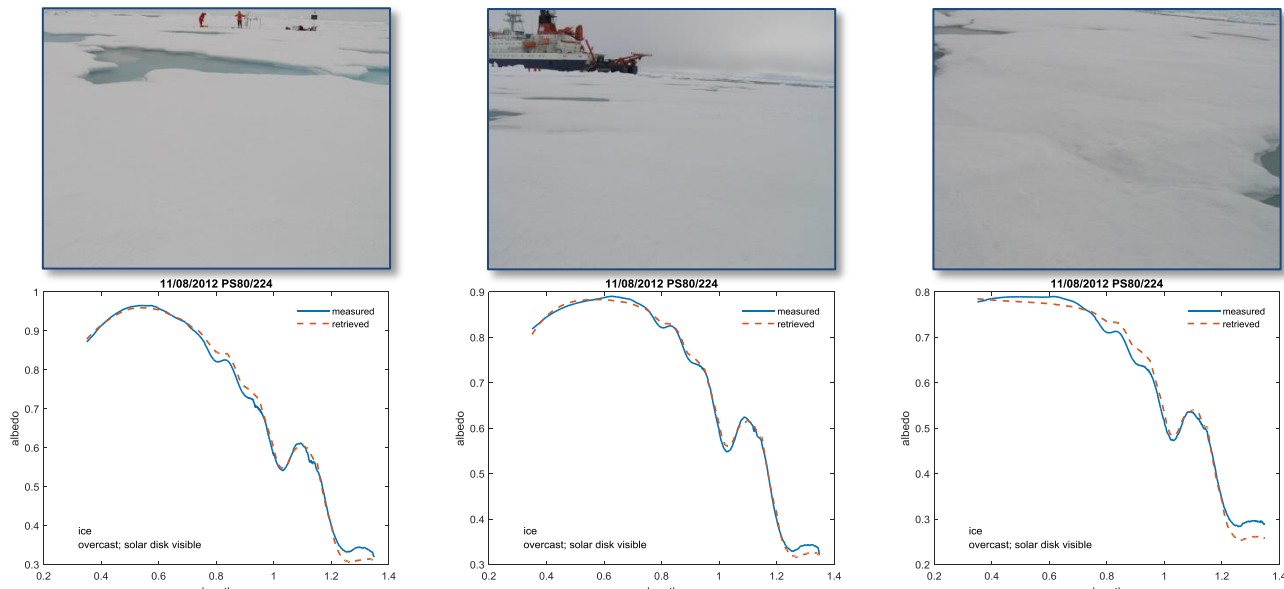

**Figure 8. Bright white ice. The surface is dry; the scattering layer is 3-8 cm thick.**





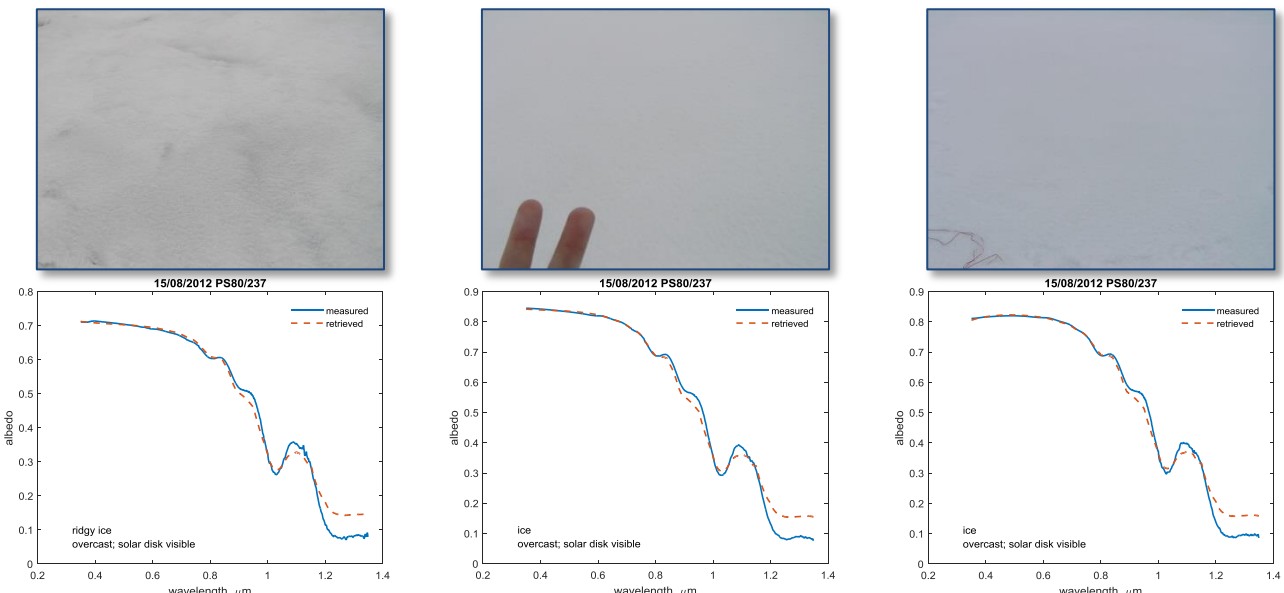

**Figure 9. Typical white ice in Arctic summer. The surface is slightly wet; the grains are larger than those in dry white ice; the scattering layer is 12-18 cm thick.**





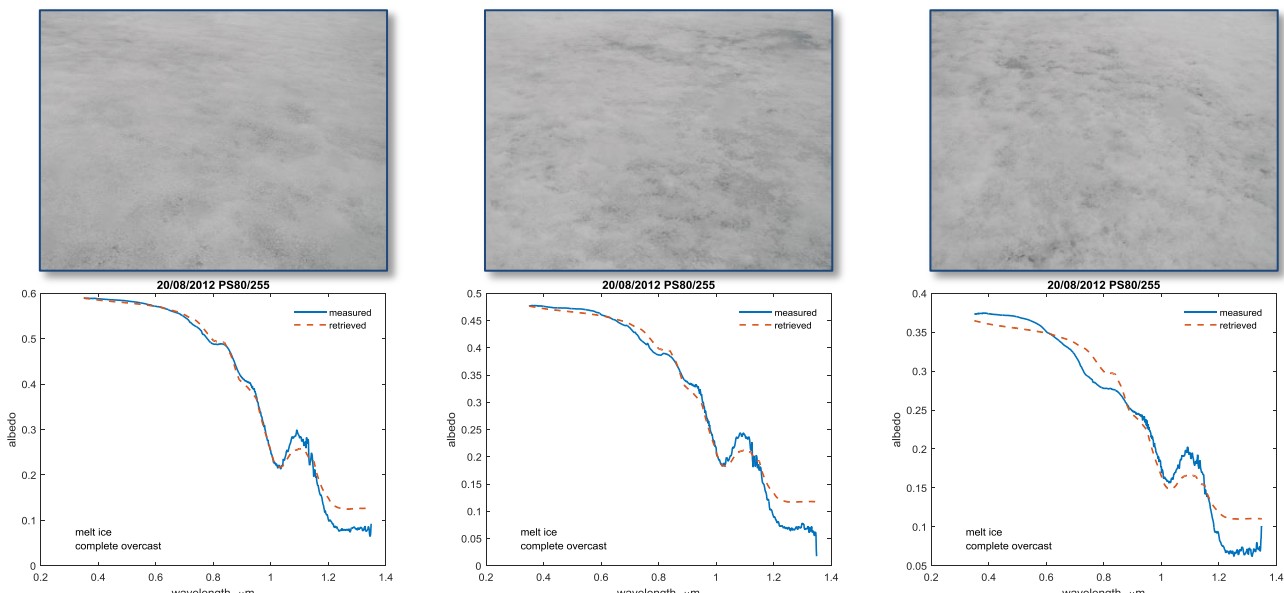

**Figure 10. Melt ice. Melting (wet) scattering layer with depth of 1.5 cm (left) and 7 cm (the others). The degree of melting increases from left to right.**



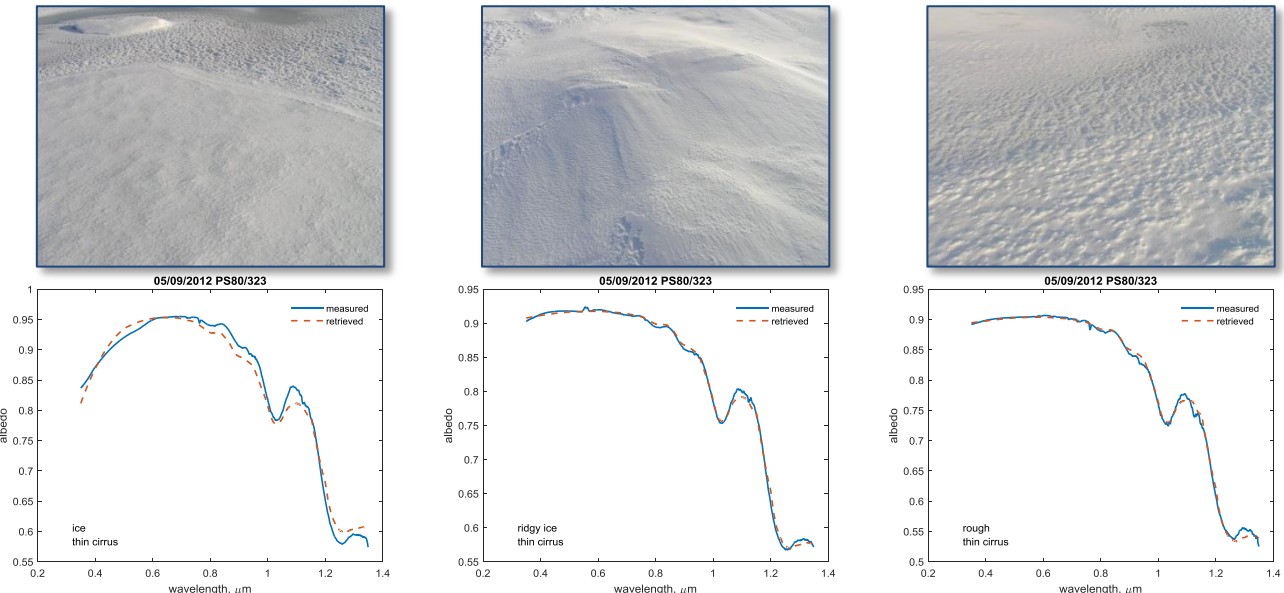

**Figure 11. Snow-covered ice. The surface is fresh fallen fine-grained snow (up to 5 cm) upon the wind crusted older scattering layer (2-4 cm).**




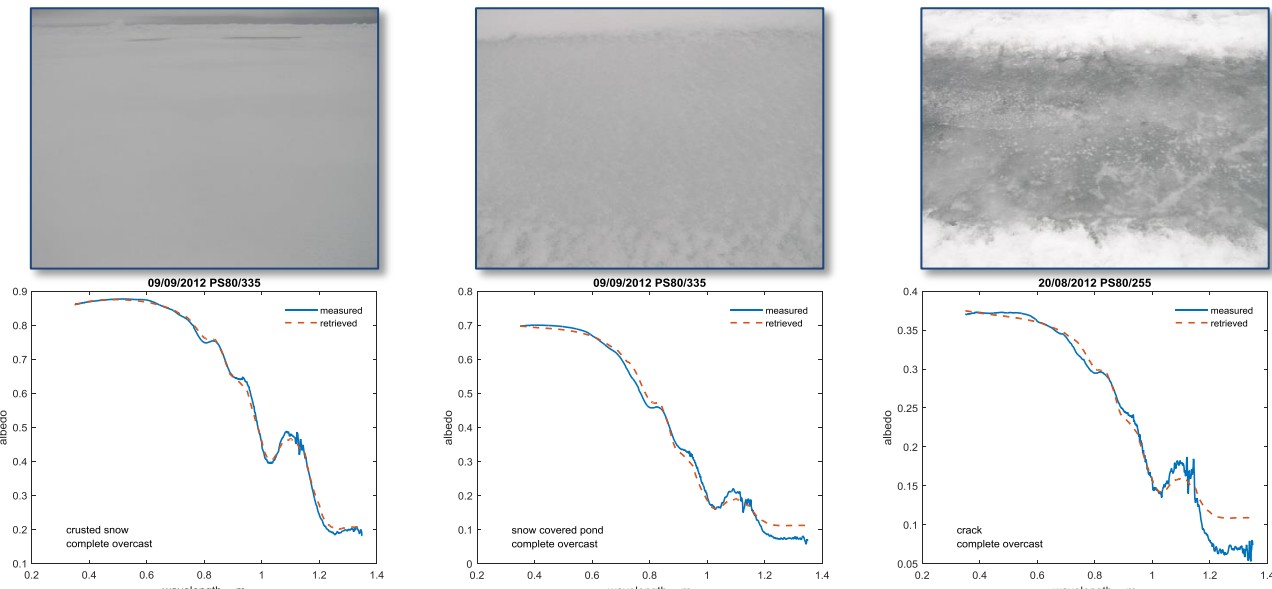

**Figure 12. Three special cases: thin wind crust on top of fine fresh snow of 4 cm thickness (left), a frozen over grey melt pond with snow on top (middle), and a frozen over crack with air bubbles and algae inclusions (right).**