# Peer review of "Reflective properties of white sea ice and snow"

_The Cryosphere, 2016_

## Referee Comment (RC1) · Anonymous Referee #1 · 21 Jul 2016

This paper is aimed at theoretical and experimental studies of reflective properties of white and snow-covered sea ice. I suggest the publication of this paper after minor corrections. The authors may address the following points: 1. I do not think that the mean photon path length (MPPL) coincides with $a_ef(p.3, line 26). Please, define MPPL and establish the link. 2. Eq. (10) and similar equations below - please, signify that they have been derived at $k = 0$. 3. Please, derive Eq. (11) 4. p.10, you show that $g = 0.67$ for your model. Do not you think that this is too small value taking into account that the scattering elements are large as compar

---

## Referee Comment (RC2) · Anonymous Referee #1 · 21 Jul 2016

This paper is aimed at theoretical and experimental studies of reflective properties of white and snow-covered sea ice. I suggest the publication of this paper after minor corrections. The authors may address the following points: 1. I do not think that the mean photon path length (MPPL) coincides with the value of a_ef. Please, establish the link. 2. Eq. (11) has been derived assuming that k=0. Please, say in this in the paper ( and also for other similar equations). 3. Please, derive Eq. (11). 4. Do not you think that the value of g=0.67 (p.10) is too small for large scatterers like ice grains?

---

## Referee Comment (RC3) · Anonymous Referee #2 · 23 Aug 2016

Referee Comments for "Reflective Properties of White and Snow-Covered Sea Ice" by Malinka et al.

General Comments

This manuscript takes an analytical approach to modeling optical properties of summer sea ice with a highly scattering surface layer. The approach is reliant upon the notion that a justified application of geometrical optics and stereology allows the use of an analytical examination of scattering, the property that dictates the optical behavior of the ice cover with the specified surface conditions (i.e. large grains of snow or drained ice). In establishing this analytical basis, the authors show that optical thickness and effective grain size can be used to determine apparent optical properties, particularly reflectance. The presented findings are relevant and useful to the sea ice community.

I recommend publication after minor revisions.

This approach is particularly useful in that simplicity is achieved with requiring only a few input parameters. Additionally, it is important to note the similarities in optical treatment of a snow cover and a summertime white ice surface scattering layer. A useful addition to the discussion or conclusion would be explicitly stated limitations to this model. Can the authors determine and optical depth and or chord length threshold at which this analytical approach no longer holds true? Maybe it is a more qualitative caveat for surface type or point during the melt season.

Specific Comments

Section 1: What about the importance with respect of larger climate models that call for absolute accuracy of 0.02 for albedo measurements (Sellers et al, 1995)? Could be worth mentioning.

Section 2 lines 23-25: What is the chord length distribution used for this mixture? What type of function? Following Malinka, 2014?

Section 2.2 lines 9-13: The value of g= 0.67 strikes me as low. However, I note the approach to obtain this value. Maybe this needs a bit of clarification and comparison to common g values for different cover types. Would this be for white ice, or snow?

Section 2.3 lines 24-26: There may be value in explaining or showing (briefly) how other contaminants could be modeled either in an additional parameter in eq. 18 or with the acknowledgement of the potential of adding absorption coefficients for Chl-a or sediment for example.

Section 3.2 line 21: The authors could add a specific example or citation to strengthen this idea.

Section 3.3 line 4: Authors can refer to Figure 3 here.

Section 4.1 lines 7-11: It would be useful to include temperature information for the
cruise and ice stations as well as the approximate thickness of the observed scattering layer on the surface of the white ice.

Technical Comments

Section 2.3 Line 4: I am not sure what '(see Figures below)' is referring to.

Figures 5-6: Add $\tau$= 8.5 and a= 3.333 mm to the figure captions.

Section 3.2 line 11 and 14: Describing variations in wavelength as 'layers' may not be optimal, particularly because line 16 and Figure 7 caption refers to the 'same layer', which I assume is an optical thickness of 8.5 and chord length 3.333 mm?

Section 5 line 8: Missing the word 'size' in phrase "effective grain size 1-4 mm".

Please also note the supplement to this comment:
http://www.the-cryosphere-discuss.net/tc-2016-153/tc-2016-153-RC3-supplement.pdf

---

## Author Comment (AC1) · 21 Sep 2016

First of all, we are grateful to our reviewers for their skilled and professional comments.

Below we will try to answer carefully to every note.

Anonymous Referee #1 This paper is aimed at theoretical and experimental studies of reflective properties of white and snow-covered sea ice. I suggest the publication of this paper after minor corrections.

Thank you.

The authors may address the following points: 1. I do not think that the mean photon path length (MPPL) coincides with the value of a_ef. Please, establish the link.

Yes, you are right. This coincidence exists only for the pure random mixture, i.e., in the

case in point. Except for the constant factor of 3/2 or 3/4 used by the different authors in the definition of aÂ■eff (depending on either radius or diameter of the equivalent sphere is used), the effective size coincides with the mean chord. On the other hand, the mean chord coincides with is the MPPL in a particle of random shape. However, for any other shape, when the angles of refraction and photon path lengths are not strictly independent (e.g. for spheres), the mean photon path length for sure does not coincide with the mean chord, which by definition assumes the random straight line field. The appropriate corrections are made in lines 25-26 on page 3.

2. Eq. (11) has been derived assuming that k=0. Please, say in this in the paper (and also for other similar equations).

In Eq. (11), as in the whole theory (Malinka, 2014), k is assumed to be small but not necessarily equal zero. In Eq.(11) you can see $\omega 0$, which can be <1, and $\alpha$, which can be of any value.

3. Please, derive Eq. (11).

Eq. (11) is derived in Malinka, 2014. To avoid misunderstandings, we put a phrase that all the Eqs. from (6) to (16) are derived there (P. 4, Ls. 5-6).

4. Do not you think that the value of g=0.67 (p.10) is too small for large scatterers like ice grains?

Note that these results are derived in the framework of the geometrical optics. The value of g with diffraction will be approximately (1+g)/2, i.e. about 0.84. The appropriate comment is added (P. 10, l. 6).

---

## Author Comment (AC2) · 21 Sep 2016

First of all, we are grateful to our reviewers for their skilled and professional comments.

Below we will try to answer carefully to every note.

Anonymous Referee #2:

General Comments: This manuscript takes an analytical approach to modeling optical properties of summer sea ice with a highly scattering surface layer. The approach is reliant upon the notion that a justified application of geometrical optics and stereology allows the use of an analytical examination of scattering, the property that dictates the optical behavior of the ice cover with the specified surface conditions (i.e. large grains

of snow or drained ice). In establishing this analytical basis, the authors show that optical thickness and effective grain size can be used to determine apparent optical properties, particularly reflectance. The presented findings are relevant and useful to the sea ice community. I recommend publication after minor revisions.

Thank you.

This approach is particularly useful in that simplicity is achieved with requiring only a few input parameters. Additionally, it is important to note the similarities in optical treatment of a snow cover and a summertime white ice surface scattering layer. A useful addition to the discussion or conclusion would be explicitly stated limitations to this model. Can the authors determine and optical depth and or chord length threshold at which this analytical approach no longer holds true? Maybe it is a more qualitative caveat for surface type or point during the melt season.

We have tried to outline the model requirements: the grain shapes should be close to random, their size should be much larger than the wavelength of light (beginning from about $10\lambda$), and the layer should be quite bright, i.e. with the albedo higher than 0.5, which means the optical depth greater than 4 (see Eq. 30). However, during the process of verification we have found (and mentioned) that "the model works quite satisfactory in the cases that stand out of the initial frames of the model." (P. 14, L. 25) We put a couple of words about the initial limitations in the Conclusion (P. 15, Ls. 4-5).

Specific Comments Section 1: What about the importance with respect of larger climate models that call for absolute accuracy of 0.02 for albedo measurements (Sellers et al, 1995)? Could be worth mentioning.

Thank you. We mentioned it (P. 2, Ls. 2-3)

Section 2 lines 23-25: What is the chord length distribution used for this mixture? What type of function? Following Malinka, 2014?

The used here chord length distribution is exponential. This has been specified immediately before Eq. (6).

Section 2.2 lines 9-13: The value of g= 0.67 strikes me as low. However, I note the approach to obtain this value. Maybe this needs a bit of clarification and comparison to common g values for different cover types. Would this be for white ice, or snow?

Regarding the g value, see Comment 4 in response to Referee #1. What concerns the type of ice/snow, we added the additional curves to the plot in Fig.3 to demonstrate the effect of grain size.

Section 2.3 lines 24-26: There may be value in explaining or showing (briefly) how other contaminants could be modeled either in an additional parameter in eq. 18 or with the acknowledgement of the potential of adding absorption coefficients for Chl-a or sediment for example.

If we understood your note right, the answer is given in the last sentence of Sec. 2.3. If the contaminant is specified and its spectrum and concentration are known, there is no problem to add its spectral absorption to the right part of Eq.Äǎ(18). Our specific choice of the yellow substance (aka CDOM) is dictated by the increase of albedo in the blue range – a fingerprint of the yellow substance. We don't see the fingerprint of Chl-A (or any other substance) in the measured spectra. As soon as we see it, we will easily include it. We inserted 'e.g. Chl-A et al.' to specify.

Section 3.2 line 21: The authors could add a specific example or citation to strengthen this idea.

Unfortunately, the discussion of this point goes far beyond the scope of the paper. We put a reference to the MPD algorithm that uses this approach, where the iteration process is described in more detail.

Section 3.3 line 4: Authors can refer to Figure 3 here.

Thanks, done.

Section 4.1 lines 7-11: It would be useful to include temperature information for the cruise and ice stations as well as the approximate thickness of the observed scattering layer on the surface of the white ice.

The scattering layer thickness is given in the figure captions. We added there the average temperature for every station.

Technical Comments Section 2.3 Line 4: I am not sure what '(see Figures below)' is referring to.

We changed to "see, e.g. Figs. 8 and 11."

Figures 5-6: Add $\tau$ = 8.5 and a= 3.333 mm to the figure captions.

Added.

Section 3.2 line 11 and 14: Describing variations in wavelength as 'layers' may not be optimal, particularly because line 16 and Figure 7 caption refers to the 'same layer', which I assume is an optical thickness of 8.5 and chord length 3.333 mm?

Thanks. We dropped out the word 'layer' to avoid confusion.

Section 5 line 8: Missing the word 'size' in phrase "effective grain size 1-4 mm".

Inserted.